

# Climate-induced phenological shift of apple trees has diverse effects on pollinators, herbivores and natural enemies

Ádám Kőrösi[1,2,*], Viktor Markó[3,*], Anikó Kovács-Hostyánszki[4], László Somay[4], Ákos Varga[3], Zoltán Elek[1], Virginie Boreux[5], Alexandra-Maria Klein[5], Rita Földesi[6] and András Báldi[4]

[1] MTA-ELTE-MTM Ecology Research Group, Budapest, Hungary
[2] Theoretical Evolutionary Ecology Group, Department of Animal Ecology and Tropical Biology, University of Würzburg, Würzburg, Germany
[3] Department of Entomology, Szent István University, Budapest, Hungary
[4] Institute of Ecology and Botany, Lendület Ecosystem Services Research Group, MTA Centre for Ecological Research, Vácrátót, Hungary
[5] Nature Conservation and Landscape Ecology, Faculty of Environment and Natural Resources, University of Freiburg, Freiburg, Germany
[6] Agroecology and Organic Farming, Institute of Crop Science and Resource Conservation, University of Bonn, Bonn, Germany
[*] These authors contributed equally to this work.

Corresponding author
Ádám Kőrösi, korozott@gmail.com

## ABSTRACT

Climate change is altering the phenology of trophically linked organisms, leading to increased asynchrony between species with unknown consequences for ecosystem services. Although phenological mismatches are reported from several ecosystems, experimental evidence for altering multiple ecosystem services is hardly available. We examined how the phenological shift of apple trees affected the abundance and diversity of pollinators, generalist and specialist herbivores and predatory arthropods. We stored potted apple trees in the greenhouse or cold store in early spring before transferring them into orchards to cause mismatches and sampled arthropods on the trees repeatedly. Assemblages of pollinators on the manipulated and control trees differed markedly, but their overall abundance was similar indicating a potential insurance effect of wild bee diversity to ensure fruit set in flower-pollinator mismatch conditions. Specialized herbivores were almost absent from manipulated trees, while less-specialized ones showed diverse responses, confirming the expectation that more specialized interactions are more vulnerable to phenological mismatch. Natural enemies also responded to shifted apple tree phenology and the abundance of their prey. While arthropod abundances either declined or increased, species diversity tended to be lower on apple trees with shifted phenology. Our study indicates novel results on the role of biodiversity and specialization in plant-insect mismatch situations.

## INTRODUCTION

There is growing evidence that in response to ongoing global climate change (*IPCC, 2014*) the phenology of functionally diverse organisms has been shifted in the last few decades (for a review see *Donnelly, Caffarra & O'Neill, 2011*). The rate of phenological shift can vary across different taxa within the same community (*Primack et al., 2009*; *Cook, Wolkovich & Parmesan, 2012*), a phenomenon also known as response diversity (*Elmqvist et al., 2003*). As a consequence, phenological overlap of interacting species can decrease, sometimes resulting in mismatches that uncouple the interaction (*Stenseth & Mysterud, 2002*; *Thackeray et al., 2010*). Examples of such phenological mismatches are known from a wide range of ecosystems, mainly in the temperate and arctic regions (*Visser & Holleman, 2001*; *Winder & Schindler, 2004*; *Post & Forchhammer, 2008*). However, our knowledge on the possible consequences of phenological mismatches on ecological interactions is still limited (*Hegland et al., 2009*; *Miller-Rushing et al., 2010*). Some studies on phenological mismatch assessed a few interacting species at two or more trophic levels (*Doi, Gordo & Katano, 2008*; *Both et al., 2009*; *Evans et al., 2013*; *Kudo & Ida, 2013*), while others examined whole ecological networks and related the change in network structure with climate warming (*Burkle, Marlin & Knight, 2013*) or quantified the rate of phenological change in many interacting species across different ecosystems (*Thackeray et al., 2010*). The consequences of mismatch in multiple interactions within a given community, however, are still largely unexplored.

Considerable attention has been devoted to mutualistic plant–pollinator networks, since earlier simulations predicted pollinator extinctions due to mismatch with food plants (*Memmott et al., 2007*). A recent review (*Forrest, 2015*) of the relationship between phenological changes and plant–pollinator interactions found that apart from a few examples of negative consequences of mismatch between plants and pollinators (*Thomson, 2010*; *Kudo & Ida, 2013*), rates of phenological advance related to global warming seem broadly consistent between generalist plants and insect pollinators at large spatial scales (*Bartomeus et al., 2011*; *Iler et al., 2013*; *Ovaskainen et al., 2013*). This consistency is likely due to the fact that insects and the plants they pollinate may use similar environmental cues to time their spring emergence (*Forrest & Thomson, 2011*), and that plant–pollinator interactions are quite flexible (*Petanidou et al., 2008*; *Benadi et al., 2014*). *Rafferty & Ives (2011)* manipulated the phenology of 14 plant species and found no temporal mismatches between flowering onset and pollinator visitation for most of them. *Burkle, Marlin & Knight (2013)* attributed a large proportion of lost plant–bee interactions in their pollination network to phenological mismatch, but could not determine whether the apparent mismatches were a cause or a consequence of pollinator declines.

Hitherto, a few studies have been able to uncover the consequences of phenological shifts for species and their trophic interactions (*Rafferty et al., 2013*). For example, recent phenological asynchrony between egg-hatching of the winter moth (*Operophtera brumata* L.) and bud burst of oak trees (*Quercus robur* L.) due to their differential response to increased spring temperature lead to natural selection, and the winter moth rapidly adapted to this environmental change resulting in recovery of synchrony (*Van Asch et al., 2007*).
Mutualistic interactions can also be disrupted when climate response is different between species. When plant phenology was experimentally manipulated, advanced flowering of *Sinapis arvensis* (L.) caused a decline in flower-visiting pollinators, but an increase of seed set (*Parsche, Fründ & Tscharntke, 2011*), while both advanced and delayed treatments led to very low reproduction of a spring ephemeral *Claytonia lanceolata* Pursh due to either frost damage or low pollinator visitation (*Gezon, Inouye & Irwin, 2016*). Nevertheless, all these studies were limited to one type of interaction, a few interacting species and/or two guilds. For future research, the importance of studies that scale up from pairwise species interactions to communities and ecosystems involving multiple trophic levels using experimental approaches has been emphasized (*Rafferty et al., 2013*).

In Europe, the apple (*Malus x domestica*) is one of the most important insect-pollinated crop plants, accounting for 16% of the EU's total economic gains attributed to insect pollination (*Leonhardt et al., 2013*). Apple orchards can harbor rich arthropod communities that largely contribute to the biodiversity and functioning of agro-ecosystems (*Rosa García & Miñarro, 2014*), while crop yield and quality strongly depend on ecosystem services, particularly pollination (*Klein et al., 2007*; *Garratt et al., 2014*) and pest control (*Cross et al., 2015*). We aimed to unravel the possible consequences of climate-induced phenological shift of apple on the abundance and diversity of arthropods at multiple trophic levels. Therefore, we manipulated the phenology of young potted apple trees in a greenhouse or a cold store and repeatedly sampled arthropod communities after transferring the trees into organic apple orchards. In this way we imitated a 'worst-case scenario' (sensu *Rafferty et al., 2013*), i.e., when climate-induced phenological shift of apple trees to earlier dates was much larger (Advanced scenario) or smaller (Delayed scenario) than that of arthropods, and thus phenological asynchrony was maximized. In the temperate zone, climate change usually advances spring phenology (*Schwartz, Ahas & Aasa, 2006*), so our Delayed scenario imitated a situation when phenology of apple trees advances to much lower degree than the phenology of arthropods. Primarily, we were interested in the response of arthropod abundance and diversity to this experimentally induced phenological asynchrony.

We hypothesized that: (i) pollinator abundance, diversity and species composition on manipulated trees will be different from the control trees (*Rafferty & Ives, 2011*; *Parsche, Fründ & Tscharntke, 2011*; *Gezon, Inouye & Irwin, 2016*); (ii) responses of herbivores will correlate with their degree of specialization to apple (measured as degree of monophagy), since more specialized interactions are expected to be more vulnerable to phenological mismatches (*Memmott et al., 2007*; *Van Asch & Visser, 2007*; *Miller-Rushing et al., 2010*); (iii) natural enemies as secondary consumers may be either less or even more affected than herbivores, depending on the response of their prey (herbivores) and how strongly they are coupled to them.

## MATERIALS & METHODS

### Experimental design and sampling

We manipulated the phenology of potted, 3 year old apple trees ($n = 182$, cv. Resi, $\sim$2.5 m height, $\sim$3 cm trunk diameter) by keeping them either in a greenhouse (advanced
**Table 1** Apple tree phenology in each of the five treatments.

| Treatment | Abbreviation | Number of trees | Date of planting outdoors | Peak flowering, pollinator sampling | Days of flowering |
|---|---|---|---|---|---|
| Advanced1 | A1 | 31 | 17.04.2013 | 19–20.04.2013 | 6 |
| Advanced2 | A2 | 31 | 19.04.2013 | 24–25.04.2013 | 6 |
| Control | C | 42 | 05.04.2013 | 2–3.05.2013 | 5 |
| Delayed1 | D1 | 39 | 30.04.2013 | 17–19.05.2013 | 6 |
| Delayed2 | D2 | 39 | 07.05.2013 | 7–8.06.2013 | 7 |

treatments), a cold store (delayed treatments), or outdoors (control) from March 2013. Before flowering, trees were buried outdoors with their pot in five treatments during April and May (Advanced1, Advanced2, Control, Delayed1, and Delayed2. See Table 1). The experiment was conducted in three organic apple orchards in Eastern Hungary (see geographical locations in Data S1). Within the three orchards, we designated altogether eight blocks (3 + 3 + 2) and we distributed the experimental apple trees among them in a way that each block contained more or less similar numbers of trees from all treatment groups. During flowering, branches of apple trees were placed out in water canisters in all blocks to enable cross-pollination for the experimental trees. For information on the flowering periods of apple cultivars that occurred in the study orchards see Table S1.

Phenology of apple trees was documented by photographs and dates of onset of main phenological phases (bud burst; onset, peak, end of flowering) were also recorded. Five randomly selected leaves were collected from each experimental tree on 18 July and leaf size was calculated from scanned digital images (*O'Neal, Landis & Isaacs, 2002*) using ImageJ 1.49 (*Schneider, Rasband & Eliceiri, 2012*) and Adobe Photoshop 8.0 (Adobe Systems, San Jose, CA, USA) software. Ripe apple fruits were collected from all experimental trees in August. Total shoot growth was measured on three randomly chosen annual shoots per tree on 22 Nov.

Pollinators were sampled on each tree twice, on two subsequent days between 9:00 a.m. and 4:00 p.m. under favourable weather conditions (>20 °C, wind speed ≤ 3 on Beaufort scale) during peak flowering of each treatment group. At both occasions, we counted the flowers and observed the pollinators landing on the flowers on each tree for 15 min. All trees were sampled by two persons at one time. European honey bees (*Apis mellifera* L.) and bumble bees (*Bombus* spp.) (Hymenoptera: Apidae) were identified without capturing, while solitary wild bees (i.e., other wild bees than *Bombus* spp. including some semi-social species; Hymenoptera: Apoidea) and hoverflies (Diptera: Syrphidae) were captured by insect nets and preserved in 70% ethanol for later identification.

Green apple aphids (*Aphis pomi* de Geer) (Hemiptera: Aphididae) were sampled three times during the summer (14 & 27 June, 12 July). Each time the shoots were counted and the proportion of young, still growing shoots was estimated on each tree. Then three growing shoots and three non-growing (old) shoots were randomly selected and aphids were counted on them. Other herbivores and natural enemies were sampled by beating the whole canopy of each tree for 15 s with a 70-cm-long stick, collecting the fallen arthropods in a 35-cm-radius beating funnel and preserving them in 50% ethanol for later identification.

Sampling was repeated once a week from 24 April until 18 July. Buds infected by the apple blossom weevil (*Anthonomus pomorum* L.) (Coleoptera: Curculionidae) were counted on all trees one week after peak flowering. These 'capped buds' have a rusty colour, never open and stay on the tree for a while.

## Data preparation

We analyzed the effects of treatments on some characteristics of apple trees, and abundance, diversity and species composition of arthropods. Apple tree flower numbers from the two pollinator sampling occasions were averaged for each tree and square root transformed, while leaf size and shoot growth were not transformed. The number of ripe apples was divided by the number of flowers for each tree to calculate fruit set. Abundance of honey bees, wild bees (incl. bumble bees) and hoverflies from the two sampling occasions was summed for each tree. Based on the shoot and aphid counts, aphid abundance was estimated for each tree at each sampling event and then $\log(x+1)$-transformed. Both the proportion of growing shoots and aphid abundance showed a sharp decline through the three sampling occasions in all treatments and the mean of the three samples was used for analysis.

Since the number of arthropods per tree per beating event was very low, we pooled the data of all samples between 8 May and 18 July for each tree. Thus we pooled the same number of samples ($n = 11$) for all treatment groups. From herbivores, we used data of apple blossom weevil, pear lace bug (*Stephanitis pyri* (Fabr.)) (Hemiptera: Tingidae) and all other phytophagous bugs (Hemiptera: Heteroptera). The highest number of adults of *A. pomorum* was found in late April, so we pooled the samples from 24 April to 18 July for this species. Only its occurrence was analyzed as this species occurred in only one orchard, where it was absent on ~75% of the trees. *A. pomorum* occurrence was zero on Delayed1 trees, so we omitted this treatment group from the analysis. Buds infected by *A. pomorum* were found only on advanced and control trees in the same orchard. Number of capped buds was $\log(x+1)$-transformed.

In order to account for natural enemies, we used the abundance of aphidophagous beetles (Coleoptera: Coccinellidae), zoophagous (incl. zoo-phytophagous) bugs (Hemiptera: Heteroptera) and spiders (Araneae). For spider abundance, juvenile and adult spiders were all counted, while diversity and species composition analyses were based on only adult spider specimens that were possible to identify at species level. We identified sufficient number of species to analyze species composition and diversity in four taxonomic groups: wild bees (incl. bumble bees), phytophagous true bugs (excluding *S. pyri* due to its overwhelming abundance), aphidophagous beetles and spiders.

## Data analysis

We used generalized linear mixed effects models (GLMM) with treatment as the fixed effect and block ID as a random factor. Response variables were the abundance, in some cases the occurrence of arthropods, flower number, leaf area, shoot growth, fruit set and the number of buds infected by *A. pomorum* (see 'Data preparation'). For some response variables, continuous variables as additional fixed terms were included in the GLMM. In

such cases, we also tested if the interaction between treatment and the continuous variable was significant. Such continuous variables were: the number of flowers for pollinator abundance; the proportion of growing shoots for aphid abundance; aphid abundance for the abundance of aphidophagous beetles and zoophagous bugs; abundance of *S. pyri* for the abundance of spiders. We hypothesized that these covariates represent the main food sources of the corresponding arthropod groups and thus would influence their abundance. Moreover, the inclusion of these covariates ensured that we could separate the effects of altered phenology and altered physiology of apple trees caused by our experimental treatment. Continuous covariates were centered. When more than one fixed term was involved, we performed an AICc-based model selection (*Burnham & Anderson, 2002*) and results of the model with the lowest AICc value are reported.

For occurrence data, we used a binomial error distribution, otherwise we applied the most appropriate error structure based on AICc values and diagnostic plots. Abundance of *S. pyri* was zero on ∼36% of apple trees and showed a right-skewed distribution on the rest of them so we fitted a model to the log-transformed non-zero abundances with a quasi-Poisson error structure. Quasi-Poisson error structure was used also for fruit set and number of capped buds (Table S1). Zero-inflated models were used if it improved model fit (*Zuur et al., 2009*).

For the community-level analysis, we calculated and plotted Rényi's diversity profile for each treatment. Common diversity indices are special cases of Rényi diversity (*Hill, 1973*), and one community can be regarded as more diverse than another only if its Rényi diversities are all higher (i.e., their diversity profiles do not intersect) (*Tóthmérész, 1995*). We also conducted two separate redundancy analyses (RDA) with constraint variables *orchard* and *treatment*. Species matrix was transformed with the Hellinger method to improve the effectiveness of these analyses in representing ecological relationships (*Legendre & Gallagher, 2001*), and significance of the constraint term was tested by a permutation test ($10^4$ permutations). A non-metric multidimensional scaling (NMDS) with Bray–Curtis distances was applied to visualize similarity among treatments in each orchard. All analyses were made using packages glmmADMB (*Skaug et al., 2015*), lme4 (*Bates et al., 2015*), MuMIn (*Barton, 2014*) and vegan (*Oksanen et al., 2017*) of R 3.4.3 statistical software (*R Core Team, 2017*). The dataset and the R code used for the analysis are in (Data S2 & Data S3).

# RESULTS

## Tree phenology and pollination success

Trees kept in the greenhouse had their flowering advanced by 6–9 days, while flowering of cool stored trees was delayed by 16–38 days compared to the control ones. Flowering period lasted for 5–7 days in all treatments (Table 1). Number of flowers was higher on control trees than in all other treatments, but there was no difference among the other treatment groups (Fig. 1A). Leaf size was larger on advanced trees and smaller on delayed trees than on control ones (Fig. 1B), while total shoot growth was not affected by treatment (Fig. 1C). Sufficient number of fruits for statistical analysis was harvested in only one orchard. Here

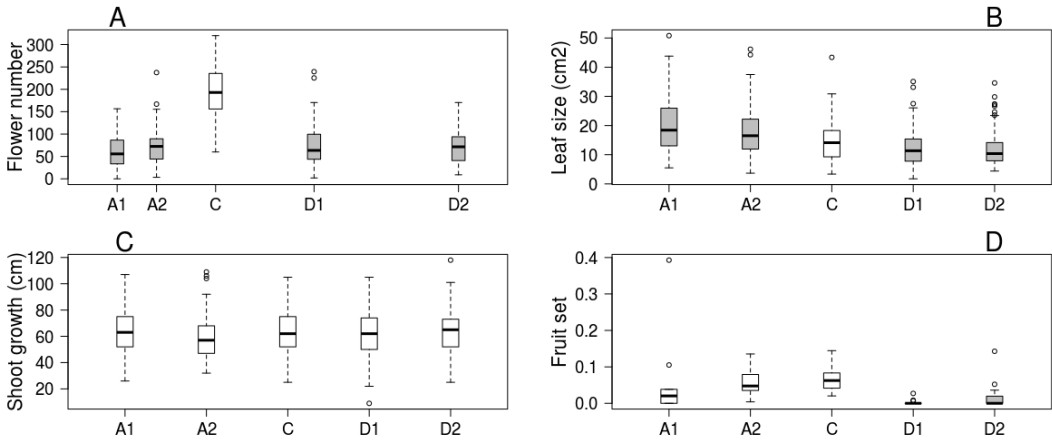

**Figure 1** **Apple tree condition in each of the five experimental treatments.** (A) Number of flowers. (B) Leaf size. (C) Shoot growth. (D) Fruit set (assessed in one orchard only, $n = 62$). Grey boxes represent treatments significantly different from the control. (A) box spacing is proportional to the time elapsed between peak flowering in each treatment.

fruit set did not differ between advanced and control trees, but delayed trees responded with lower reproductive success (Fig. 1D, Table 2 & Table S2).

## Pollinators

Flower number of apple trees had a positive effect on the abundance of all pollinator groups, but its interaction with treatment was not significant (Table 2 & Table S2). This means that the relationship between flower number and pollinator abundance had the same strength in each treatment. In other words, pollinator abundance was best explained by treatment effect plus the effect of flower number. However, since the number of flowers was not equal among treatment groups (see above), model estimates on treatment effects should be interpreted as if flower number were the same in all treatments. Model estimates of honey bee abundance were higher on delayed trees and Advanced2 trees than on control ones. Observed honey bee abundance was not higher on Advanced2 trees than on control ones (Fig. 2), but the model estimated that it would be higher if the number of flowers were equal in these two groups. Hoverfly abundance was also higher on delayed trees than on control ones. Abundance of wild bees was higher on Advanced1, Delayed1 and Delayed2 trees than on control ones (Fig. 2) (Table 2 & Table S2). To compare with other studies, we also calculated and plotted the visitation rate as the number of pollinators/15 min/1,000 flowers in each treatment group (Fig. S1).

Altogether 39 wild bee species were observed on the apple trees (Table S3). Diversity was higher on Delayed2 trees than on control and Advanced1 trees, while diversity on Delayed1 trees was lower than on all aforementioned treatments (Fig. 3A). A few species (*Andrena haemorrhoa* Fabr., *A. varians* Kirby, *Lasioglossum calceatum* Scopoli, *Osmia cornuta* Latr.) occurred in high abundances on Advanced1 trees, but in much lower abundances in all other treatments, resulting in the significantly higher abundance on Advanced1 trees. RDA revealed that *treatment* had an effect on species composition of wild bee assemblages and it

**Table 2  Parameter estimates (SE) of the best models for each response variable.** Control was the reference level of treatment in all models, significant terms are bold. Red upward and blue downward arrows indicate that the response variable was significantly higher or lower, respectively, in the given treatment than in the control group. See Table S2 for full model outputs. Diversity was compared using Rényi's diversity profiles (Fig. 3), thus no parameter estimates are available. Arrows indicate that diversity profiles were below (blue) or above (red) that of the control group.

| | | Advanced1 | Advanced2 | Delayed1 | Delayed2 |
|---|---|---|---|---|---|
| Tree condition | Flower number | **−6.35 (0.64)** ↓ | **−5.64 (0.63)** ↓ | **−5.71 (0.59)** ↓ | **−5.63 (0.59)** ↓ |
| | Fruit set | −0.04 (0.42) | −0.03 (0.41) | **−2.95 (1.27)** ↓ | **−1.18 (0.58)** ↓ |
| | Leaf area | **5.7 (0.67)** ↑ | **3.2 (0.67)** ↑ | **−2.4 (0.63)** ↓ | **−2.8 (0.63)** ↓ |
| | Shoot growth | 0.38 (2.70) | −3.8 (2.70) | −1.66 (2.53) | −0.44 (2.53) |
| Pollinators | Honey bee abundance[a] | −0.34 (0.23) | **0.47 (0.17)** ↑ | **1.13 (0.15)** ↑ | **1.89 (0.14)** ↑ |
| | Hoverfly abundance[a] | −0.20 (0.48) | −0.02 (0.43) | **1.46 (0.31)** ↑ | **2.46 (0.29)** ↑ |
| | Wild bee abundance[a] | **2.03 (0.23)** ↑ | 0.16 (0.29) | **0.81 (0.24)** ↑ | **0.67 (0.26)** ↑ |
| Herbivores | *A. pomorum* occurrence | −1.67 (0.96) | −1.67 (0.96) | ↓[b] | **−1.99 (0.94)** ↓ |
| | *A. pomorum* infected buds | **−1.09 (0.49)** ↓ | −0.13 (0.35) | ↓[b] | ↓[b] |
| | Aphid abundance[a] | 0.43 (0.30) | 0.15 (0.34) | **0.77 (0.19)** ↑ | −1.32 (0.29) ↑ |
| | *S. pyri* abundance | −0.15 (0.11) | 0.04 (0.11) | **−0.26 (0.11)** ↓ | **−0.33 (0.11)** ↓ |
| | Phytophagous bug abundance | −0.42 (0.24) | 0.01 (0.22) | −0.29 (0.22) | −0.22 (0.21) |
| Natural enemies | Aphidophagous beetle abundance[a] | −0.15 (0.23) | **−0.54 (0.26)** ↓ | 0.36 (0.26) | **0.99 (0.26)** ↑ |
| | Zoophagous bug abundance | 0.84 (0.57) | 0.73 (0.57) | **1.72 (0.51)** ↑ | **1.74 (0.51)** ↑ |
| | Spider abundance[a] | −0.16 (0.14) | −0.104 (0.14) | −0.09 (0.13) | **−0.57 (0.15)** ↓ |
| Diversity | Wild bees | | | ↓ | ↑ |
| | Phytophagous bugs | | | ↓ | ↓ |
| | Aphidophagous beetles | | | | ↓ |
| | Spiders | ↓ | ↓ | ↓ | ↓ |

**Notes.**
[a] Indicates that an additional covariate also had significant effect on the response variable (number of flowers for pollinators, PGS for aphids and prey abundance for natural enemies; see text).
[b] Indicates that the response variable in a given treatment was zero on all trees.

explained 12.3% of total variation, while *orchard* explained only 3.4% (Table S4). Samples of treatment groups were separated on the NMDS plot (Fig. S2).

## Herbivores

Both the occurrence of *A. pomorum* and the number of capped buds were lower in all treatments than in control. The difference was significant on Delayed2 trees (occurrence) (Fig. 4A), and on Advanced1 trees (capped buds) (Table 2). We found an interaction between phenological treatment and proportion of young growing shoots (PGS) in their effects on aphid (*A. pomi*) abundance, as it was positively related to PGS in the delayed, but not in the other treatments (Fig. 4B & Fig. S3, Table 2 & Table S2). However, both PGS and aphid abundance were higher on delayed trees. According to this model, the higher aphid abundance on delayed trees is a consequence of the higher PGS on them. Non-zero abundance of *S. pyri* was lower on delayed trees than on control ones (Fig. 4C). Finally,

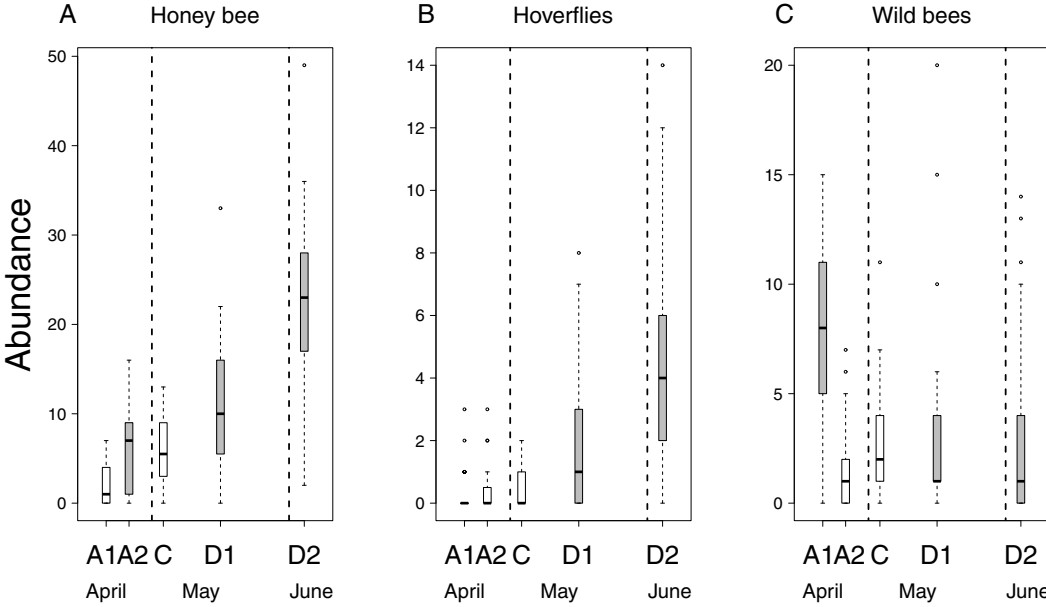

**Figure 2 Abundance of pollinators in each of the five experimental treatments.** Box spacing is proportional to the time elapsed between peak flowering in each treatment. Grey boxes represent those treatments where parameter estimates were significantly different from the control, based on the best models that included the number of flowers as well (see text). Note that this does not mean that pollinator abundance was significantly different in these treatments. Dashed lines indicate the beginning and end of May. (A) Abundance of honey bees. (B) Abundance of hoverflies. (C) Abundance of wild bees.

the abundance of phytophagous bugs was unaffected by treatment of apple trees (Table 2 & Table S2).

A total of 30 phytophagous bug species (excluding *S. pyri*) were identified in the samples (Table S3). Phytophagous bug diversity was lower on delayed than on control and advanced trees (Fig. 3B). RDA revealed no effect of *treatment* on species composition, while *orchard* had a significant effect and explained 11.2% of the total variation (Table S4, Fig. S2).

## Natural enemies

Abundance of aphidophagous beetles was higher on Delayed2 trees and lower on Advanced2 trees than on control ones (Fig. 4D). Beetle abundance was negatively related to aphid abundance (Table 2 & Table S2, Fig. S4). *Harmonia axyridis* Pallas (harlequin ladybird) was the dominant species (56% of all aphidophagous beetles). When its data were analyzed separately, we found higher abundance of *H. axyridis* on Delayed2 trees than on control ones, while abundance of all other species was unaffected by treatment (Fig. S5). Abundance of zoophagous true bugs was higher on delayed trees than on control ones (Fig. 4E), but aphid abundance had no effect on it. Spider abundance was lower on Delayed2 trees than on control trees if only treatment was included as a predictor (Fig. 4F). However, in the best model the abundance of *S. pyri* was also included as a covariate, and its interaction with treatment was significant: spider abundance increased with the abundance of *S. pyri* in Advanced2 and Control treatments (Table 2 & Table S2, Fig. S6).

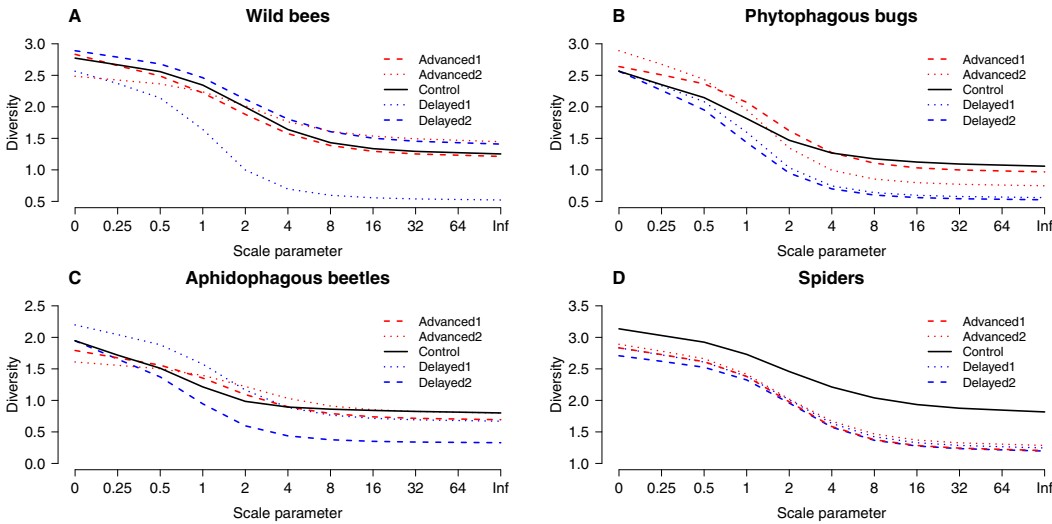

**Figure 3** "Rényi" diversity profiles of wild bees (A), phytophagous bugs (B), aphidophagous beetles (C) and spiders (D) in each of the five experimental treatments along the scale parameter (A). $a = 0$: log(species richness); $a = 1$: Shannon–Wiener index; $a = 2$: Simpson index.

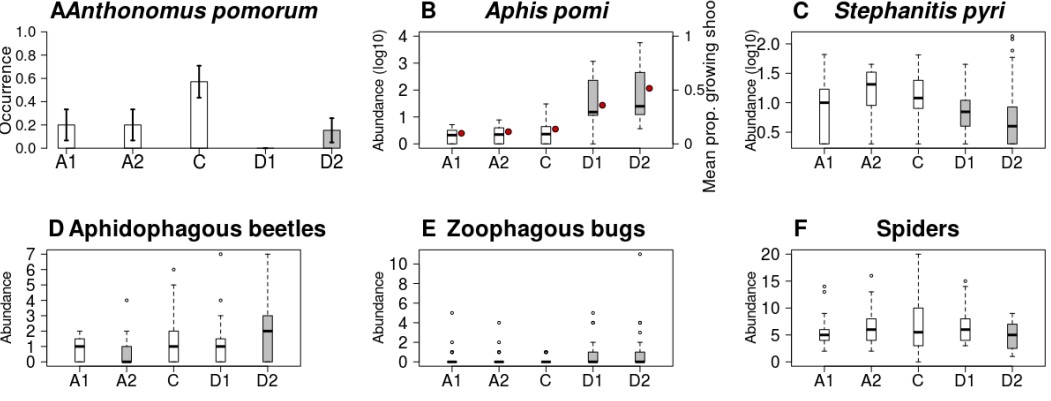

**Figure 4** **Abundance/occurrence (mean ± SE) of herbivores (A–C) and natural enemies (D–F) in each of the five experimental treatments.** Grey bars/boxes represent treatments significantly different from the control. Red dots on (B) show mean proportion of growing shoots. (A) *A. pomorum* was found in one orchard only ($n = 47$). (C) only non-zero data are shown ($n = 116$).

A total of 12 aphidophagous beetle and 40 spider species were identified in the samples (Table S3). Diversity of aphidophagous beetles tended to be lower on Delayed2 trees than on control and Delayed1 trees (Fig. 3C). Higher abundance and lower diversity on Delayed2 trees was due to the outstanding number of *H. axyridis*. *Treatment* had a significant effect on species composition, but it explained only 4.13% of total variation, while *orchard* was not significant. Diversity of spiders was lower in all treatments than on control trees

(Fig. 3D). Species composition was not affected by *treatment*, while *orchard* explained only 3.5% of total variance (Table S4, Fig. S2).

## DISCUSSION

### Tree condition and pollinators

Food availability is an important factor that governs the activity and population density of pollinating species (*Westphal, Steffan-Dewenter & Tscharntke, 2003*; *Steffan-Dewenter & Schiele, 2008*). This is also illustrated by the increase of pollinator abundance with the number of apple flowers in our study. Altered flower abundance and flowering time and non-efficient flower-visitation due to climate change are therefore among the most important potential threats to fruit set of apple trees. In our experiment, control trees had on average about three times more flowers than treated ones indicating that altered temperature implied a physiological stress and/or changed the resource allocation within a tree (*Bos et al., 2007*).

The earliest flowering trees were mostly visited by wild bees, while honey bees and hoverflies occurred in lower numbers. The outstanding number of wild bees on Advanced1 trees in mid-April was due to a few dominant species that start foraging at lower temperatures in early spring (*Torchio, 1991*). These species might have accumulated on early-flowering apple trees, due to the lack of alternative floral resources in the landscape (*Moise & Henry, 2010*). Later on the abundance of honey bees and hoverflies showed an increase during the season that might provide suitable flower visitation for the delayed apple tree flowers too. Flowering of the control trees coincided with that of all other apple trees in the orchards that may have caused a dilution effect as pollinators must have been strongly attracted by other trees as well (*Mitchell et al., 2009*; *Kovács-Hostyánszki et al., 2013*; *Riedinger et al., 2014*).

Species composition of wild bee assemblages visiting the apple trees profoundly changed during the ∼7 weeks of the flowering period of manipulated trees. Earliest flowering trees were visited by a few dominant species, but assemblages on trees flowering later were more even and diverse. These correspond to the results of *Rafferty & Ives (2012)* who manipulated the phenology of two perennial forbs and observed changes in species composition of flower-visiting wild bees during five weeks of flowering. We can also conclude that high diversity of wild bees can ensure phenological synchrony with apple tree flowering due to complementarity among bee species' activity periods and to differential responses among bee species to warming (*Bartomeus et al., 2013*). Wild bees, however, can have an outstanding importance in apple tree pollination in the case of advanced flowering, while honey bees and hoverflies may ensure efficient pollination in the case of the delayed scenario.

Efficient flower visitation mainly by wild bees on the advanced apple trees was verified by their similar fruit set to control trees. However, absolute number and total biomass of the fruits were higher on control trees (Fig. S7), which suggests that the manipulated trees might have allocated their resources to survival rather than to reproduction (*Barboza, Parker & Hume, 2009*). Fruit set was lower on delayed trees than in other treatments. Possibly,

delayed trees were not limited by pollinators, but rather late transfer from cold store to the gardens delayed their development causing a higher fruit abortion (*Bos et al., 2007*).

Recent studies suggest that pollination success of apple trees responds positively to species richness of wild bees (*Mallinger & Gratton, 2015*; *Földesi et al., 2016*). If we use visitation frequency as a surrogate of pollination service (*Vazquez, Morris & Jordano, 2005*) then we can conclude that trees with manipulated phenology were not limited by pollinator availability. However, in agricultural landscapes with less semi-natural habitats the number of wild bees can be very low and apple pollination may fully depend on honey bees (*Garibaldi et al., 2011*; *Burkle, Marlin & Knight, 2013*). In such cases, apple trees with an advanced phenology may lack sufficient pollination, and climate change could have profound effect on apple yield. Thus our results confirm that biodiversity can be crucial for the longer-term resilience of ecosystem services (*Oliver et al., 2015*).

## Herbivores

The specialist *A. pomorum* was almost absent and caused no damage on delayed trees, and it caused less damage on Advanced1 trees than on control ones. This beetle lays eggs in the flower buds, so its phenology must be highly synchronized with that of apple trees. The phenological shift of apple trees had the most adverse effects on this species among the herbivores assessed in our experiment. Aphids prefer the high water and sugar content of growing plant shoots (*Stoeckli, Mody & Dorn, 2008*) and population size of the less-specialized oligophagous aphid, *A. pomi* can rapidly increase by each generation usually reaching a peak in June—early July in apple orchards (*Markó et al., 2013*; *Nagy, Cross & Markó, 2013*). In our study, *A. pomi* showed peak abundance in mid-June, when delayed trees were still in their early and intensive growing phase providing a highly favorable food source. Total shoot growth of apple trees was similar, but its timing was very different among treatments, and it was beneficial for aphids in the delayed treatments. We note that not all aphid colonies were identified to species level in the study. We identified a random subsample of the aphids and the vast majority of them were *A. pomi*. Moreover, *A. pomi* is usually the dominant *Aphis* species (>80%) and much more abundant than *A. spiraecola* (Patch) in apple orchards in Hungary (*Borbély et al., 2017*). The even less specialized bug *S. pyri*, which is an oligophagous pest of apple feeding on Rosaceae (*Wachmann, Melber & Deckert, 2006*), was less abundant on delayed than on control trees, which resulted from the absence of the first generation (in May) from trees with delayed phenology (Fig. S8). Finally, the absolutely unspecific herbivore group of phytophagous bugs was unaffected by the phenology of apple trees. These results together support the theoretical expectations that predict a positive correlation between the degree of specialization of ecological interactions and their sensitivity to phenological mismatches (*Memmott et al., 2007*; *Miller-Rushing et al., 2010*). From the perspective of resource-consumer dynamics, the highly diverse responses of oligophagous and generalist herbivores that we found to the altered phenology of apple trees are in line with recent theoretical models predicting that changes in phenology alone can lead to qualitatively different dynamics of consumers according to their life-history (*Bewick et al., 2016*).

### Natural enemies

Natural enemies and their prey pests in agro-ecosystems provide an example of predator–prey relationships between primary and secondary consumers. In our study, abundances of natural enemies were affected by apple tree phenology and also related to prey abundances. Although we found some bottom-up effects of prey abundance, these were quite weak and natural enemies were more affected by the phenology of apple trees. For example, ladybirds were coupled to aphids, but also affected by apple tree phenology. The two delayed treatments resulted in similarly high aphid abundances, but Delayed1 trees were not visited by higher number of ladybirds than control trees, while Delayed2 trees were strongly invaded by *H. axyridis*. This is probably because peak aphid abundance on Delayed2 trees coincided with the emergence of the first generation of *H. axyridis* (*Honek et al., 2018*). The higher number of zoophagous bugs on delayed trees and the lack of statistical relationship with aphid abundance are likely due to the fact that the most dominant zoo-phytophagous mullein bug *Campylomma verbasci* (Meyer-Dür) (∼60% of all zoophagous true bugs) occurred much before the aphid peak (in late May) and occupied mainly the delayed trees. Finally, spider abundance in two treatment groups was related to the abundance of *S. pyri* which is probably an important prey of spiders (*Bogya, Markó & Szinetár, 2000*). Thus spiders were presumably indirectly affected by the phenological shift of apple trees via altered prey abundance, but only on control trees and Advanced2 trees the phenology of which was the least shifted compared to control trees. In delayed treatments, with much larger shifts, the effect of prey abundance was negligible. Due to the delayed phenology of these trees, spiders avoided them in May when their abundance usually shows a peak (*Markó et al., 2009*) (Fig. S8).

## CONCLUSIONS

Our results indicate that spatio-temporal variation in environmental conditions may play an important, taxon-specific role in the responses to climate-induced phenological asynchrony. For instance, wild bees showed virtually no spatial variation in their responses to the phenological shift of apple (Table S2), but they exhibited a profound change in species composition during the ∼7 weeks long flowering period. In contrast, a large amount of variation in herbivore abundance was explained by the random term indicating a high spatial variation, i.e., large differences both between and within orchards. Species composition of bees and aphidophagous beetles, which are relatively mobile, was rather affected by the phenology of apple trees, while in case of less mobile phytophagous bugs and spiders, variation was higher between orchards than between treatments. These suggest that responses to climate-induced phenological changes of interacting species can be scale-dependent.

We also highlight that species diversity was higher on manipulated trees than on control ones in only one case, otherwise the phenological shift of apple trees always led to lower (or similar) species diversity. The causes of altered diversity are taxon-specific. In case of spiders and phytophagous bugs both the abundance and species richness were higher on the control trees and only a subset of those species were found on manipulated trees. For wild

bees and aphidophagous beetles, the dominance of one or a few species on manipulated trees reduced diversity. We conclude that altered phenology of apple trees led to a few winners—many losers situation: it was beneficial for a few arthropod species, but rather unfavorable for most of them.

In general, both the abundance and diversity of arthropods were more strongly affected by delayed than by advanced phenology of apple trees. This may be a consequence of larger shifts in the phenology of delayed compared to advanced trees, as we accounted for altered tree physiology in our analyses. The larger phenological shift of delayed trees was due to the fact that cold store extended the dormancy of apple trees unexpectedly long. Delayed2 trees stayed only one week longer in the cold store than Delayed1 trees, but they needed ca. 30 days from planting until peak flowering, while Delayed1 trees needed only 16 days. We admit that such large phenological mismatches are slightly unrealistic in the current scenarios of climate change, so the results on delayed trees should be interpreted cautiously.

Arthropod abundance was either lower or higher, while diversity was rather similar or lower on the manipulated apple trees than on control ones. In line with our hypotheses, species composition of pollinator assemblages differed among treatment groups, and the most specialized herbivore was negatively affected by all treatments, while less specialized herbivores were affected (either positively or negatively) by only the more shifted delayed treatment. Our study present clear evidence that climate-induced phenological mismatch between an orchard crop and multiple groups of arthropods can have diverse effects on abundance, diversity and species composition of arthropods. Consequences of these changes on ecosystem services should be addressed by further research.

## ACKNOWLEDGEMENTS

We are indebted to Zsolt Józan and Balázs Keresztes for the identification of bees and spiders, respectively. We are grateful to the farmers for cooperation and to Lídia Homolya for her assistance in the laboratory work. We used free softwares: ImageJ, LibreOffice, R, RKWard and Xubuntu.

### Funding

This study was supported by the Hungarian Scientific Research Fund OTKA 101940 and 'Lendület' project of the Hungarian Academy of Sciences. Anikó Kovács-Hostyánszki (AK-H) was a Bolyai and MTA Postdoctoral Fellow. Virginie Boreux (VB) and Alexandra-Maria Klein (AMK) were supported by BiodivERsA2014-74 "EcoFruit". The funders had no role in study design, data collection and analysis, decision to publish, or preparation of the manuscript.

### Grant Disclosures

The following grant information was disclosed by the authors:
Hungarian Scientific Research Fund OTKA: 101940.

Hungarian Academy of Sciences.
EcoFruit: ERsA2014-74.

## Competing Interests

The authors declare there are no competing interests.

## Author Contributions

- Ádám Kőrösi conceived and designed the experiments, performed the experiments, analyzed the data, prepared figures and/or tables, authored or reviewed drafts of the paper, approved the final draft.
- Viktor Markó conceived and designed the experiments, performed the experiments, analyzed the data, authored or reviewed drafts of the paper, approved the final draft.
- Anikó Kovács-Hostyánszki conceived and designed the experiments, performed the experiments, authored or reviewed drafts of the paper, approved the final draft.
- László Somay and Rita Földesi conceived and designed the experiments, performed the experiments.
- Ákos Varga performed the experiments.
- Zoltán Elek conceived and designed the experiments, performed the experiments, authored or reviewed drafts of the paper.
- Virginie Boreux authored or reviewed drafts of the paper.
- Alexandra-Maria Klein conceived and designed the experiments, analyzed the data, authored or reviewed drafts of the paper, approved the final draft.
- András Báldi conceived and designed the experiments, authored or reviewed drafts of the paper, approved the final draft.

## Data Availability

The raw data and R code are provided as Supplemental Files.

## Supplemental Information

Supplemental information for this article can be found online at http://dx.doi.org/10.7717/peerj.5269#supplemental-information.

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
