# Peer review of "Climate-induced phenological shift of apple trees has diverse effects on pollinators, herbivores and natural enemies"

_PeerJ, doi:10.7717/peerj.5269_

## Round 0.1 · original submission · Minor Revisions

I enjoyed reading this well-conducted, well-written, and interesting paper. I look forward to seeing it published in PeerJ, but would like to urge you to carefully address the reviewers' comments.

Especially reviewer 1 lists a number of issues that need attention, e.g. regarding the figure design (boxplots instead of barplots...), the supplementary material, and the discussion and interpretation of the findings.

I kindly ask you to thoroughly revise your paper by implementing these and also the other comments by both reviewers.

Reviewer 1 ·

Basic reporting

The present study aimed to evaluate potential effects of climate change on the ecological interaction between apple and different animal groups: pollinators, pests and natural enemies. This is an interesting topic and research is welcomed.

The background helps to fix the aims of the MS and shows relevant literature. (L90. The correct reference is Rosa García & Miñarro 2014; the same in the reference list).

Bar graphs were used to show the results in Figures 1 to 4. However, bar graphs do not provide information about the distribution of the data and are not the most appropriated for continuous data (Weissgerber et al 2015 PLOS Biology 13 (4)). Boxplots could be a better choice. The authors used grey colors when treatments differed from the control. It is very visual but the common method of using different letters for different values could be more informative at showing differences among treatments.

The y-axis legend of Figure 2, ‘Abundance’ could be also more informative (e.g. number/1000 flowers/min), for an easier comparison of visit rate with other studies.

Legend of Figure 3. Is it necessary to provide: ‘a = 0: log (…) Simpson index’?

Supplemental Figures are provided as .eps files which could be difficult to open. When I finally managed to open them I could not find figure legends. I suggest to provide them in a more common way.

Raw data are supplied.

Experimental design

The authors use a clever method to manipulate bud burst: a greenhouse to advance phenology and a cold store to delay it. However, whereas the greenhouse advanced the bloom peak a reasonably period (6-9 days), the cold store retarded it from 16 up to 38 days, which is a huge delay in any current scenarios of climate change. Thus, this fact partially conditions the results, in particular those of treatment Delayed2. A much lower difference in days would be a more realistic scenario.

In addition, and as authors pointed out and tried to explore (they recorded lower flower number, leaf size and fruit-set in delayed trees), such methodology might have had physiological consequences for trees that had affected the plant-animal relationships.

These two points should be more deeply reflected in the interpretation of the results.

I think it is not clear enough how pollinators were sampled, if all the tree was observed, how many trees, or if transects were done… I understand that on each treatment pollinators were sampled two days, 15 min each day (i.e. 30 minutes in total) (I have a concern related with that and with the results and their interpretation. Wild bees were much more abundant in treatment A1 than in the rest, and the authors argued that it is because earlier species occurred at that moment (‘due to the lack of alternative floral resources’) and that later (control) they diminished due to a dilution effect of having many more flowers. However, three days later (A2) those wild bees were less than 8 times lower (but conditions (e.g alternative resources) should not have changed so much), and then increased again in the delayed treatments (similar results in diversity; Fig 3A). I wonder if this result could be the result of a sampling effect because pollinator assemblages may change considerably according to the time of the day as well as with the weather conditions, and two observations could not be enough to reflect the pollinator assemblage if conditions are not exactly the same. Detailed information on pollinator sampling and on time of the day and weather conditions should be provided, and a potential sampling effect should be also considered at interpreting the results.

L115. What is the phenology of cv Resi, in comparison with reference cvs (e.g. Golden Delicious)?

L117-118. I do not understand that of eight blocks and three orchards.

L121. ‘Similar numbers’ were used but see Table 1 (from 31 to 42 trees).

L143-146. Beating started before trees in the delayed treatment had flowered. How could this have affected the results?

L164-165. I do not understand the reasoning.

Validity of the findings

Results are in general valid but comments above should be considered.

Is the Conclusions section too long?

Additional comments

L285-286. And likely of blossom weevils.

L330. Could the higher number of flowers in the control partially explain the higher abundance of weevils?

L332-333. Do you mean more adverse effects than in other herbivores? Then, please rewrite.

L335. Is A. pomi a low-specialized oligophagous aphid and not a specialized one that spend all the cycle in this host? Are you sure that all they were Aphis pomi and not the similar species A. spiraecola (also frequent in Hungarian orchards: Andreev 2007. Journal of Plant Protection Research, 47, 109–112.)?

Regarding green aphids and its relation with its host, apple, it could be also discussed that delayed phenology could hamper aphids to develop from winter eggs as those hatch before bud burst and fundatrices would die by lack of food, as happens with the rosy apple aphid (Miñarro and Dapena 2007. Environmental Entomology 36(5): 1206-1211). Despite that could not be reflected in the present study it is likely to happen in natural conditions. Although it is also true that secondary colonization is possible in green aphids, unlike rosy aphid.

·

Basic reporting

The authors studied the possible effects of climate change, here through shift in the phenology of apple cultivars, on pollinisation and abundance of some pests and their natural enemies. This kind of approach is novel for agricultural conditions. The general context, and the associated concepts, are well and clearly presented in the Introduction section making it easy to read and very interesting.

Experimental design

The experimental design is well introduced (worst-case scenario) and is scientifically sound (and the number of replicates impressive). I have only minor comments on that point : (i) the date of outdoor planting are not ranked according to the different treatments (is it normal?) and (ii) the effects of the orchard where the experiment was carried out (and of its phenology) is only marginally presented and will appreciate to find a more in-depth discussion about the possible effects (bias?) of this on the main results.
Moreover, it was strange not to find more information on the apple cultivars used (for the experiment (indoor previously) and for the orchard site).

Validity of the findings

The Discussion is balanced (between clear findings and others requiring some speculation) and well constructed.

Additional comments

Other comments :
L76: the word selection here is too vague (selction of what)
L118 n=8 for each block, right?
L115: provide information on the apple cultivars
L137: something unclear: how to monitor the pollinators for 15 mn if an insect is sometimes used (it is a perturbation)
L139: no information on the rosy apple aphids (presence and infestation rate)?
L158: « thus » sounds strange here (a sharp decline do not prevent the use of raw data in a mixed model)
L180: it is very surprising that the orchard was not a random factor in this model!!!
L189-191: better explain (I do not understand)
L221: if only one orchard was considered, this important information should be provided in the figure legend
L233: hard to follow (especially because visually the mean values are close but then significantly different). Why not dividing the mean values by the number of flowers?
L391-394: it could mean that evenness would be modified. Then the absence of this index (due to Renyi) is strange and disappointing
Figure 1: not sure ‘fitness’ is the right word (fitness is the number of descendants per generation) and for me leaf size is not directly associated to fitness.
Figure 1- panel D: the SD (or SE) is sharply decreasing between advanced 1 and Control. Can this be discussed?
Figure 3: eveness is missing
Figure4: why not separating spider into families, those than can be known to be natural enemies in apple orchards and those with unknown or even negative effects (for example those known to predate syrphids)?
Table 2: (last row) does it mean that spiders are very sensitive to phenological shifts? and then why?

---

## Round 0.2 · accepted · Accept

You have done a very nice job in addressing and - where necessary - implementing the reviewers' comments. Thank you very much! I look forward to seeing your manuscript published.

Best wishes,
Christian

#